# Bayesian entropy estimation for binary spike train data using parametric prior knowledge

**Evan Archer**[13], **Il Memming Park**[123], **Jonathan W. Pillow**[123]
1. Center for Perceptual Systems, 2. Dept. of Psychology,
3. Division of Statistics & Scientific Computation
The University of Texas at Austin
{memming@austin., earcher@, pillow@mail.} utexas.edu

## Abstract

Shannon's entropy is a basic quantity in information theory, and a fundamental building block for the analysis of neural codes. Estimating the entropy of a discrete distribution from samples is an important and difficult problem that has received considerable attention in statistics and theoretical neuroscience. However, neural responses have characteristic statistical structure that generic entropy estimators fail to exploit. For example, existing Bayesian entropy estimators make the naive assumption that all spike words are equally likely *a priori*, which makes for an inefficient allocation of prior probability mass in cases where spikes are sparse. Here we develop Bayesian estimators for the entropy of binary spike trains using priors designed to flexibly exploit the statistical structure of simultaneously-recorded spike responses. We define two prior distributions over spike words using mixtures of Dirichlet distributions centered on simple parametric models. The parametric model captures high-level statistical features of the data, such as the average spike count in a spike word, which allows the posterior over entropy to concentrate more rapidly than with standard estimators (e.g., in cases where the probability of spiking differs strongly from 0.5). Conversely, the Dirichlet distributions assign prior mass to distributions far from the parametric model, ensuring consistent estimates for arbitrary distributions. We devise a compact representation of the data and prior that allow for computationally efficient implementations of Bayesian least squares and empirical Bayes entropy estimators with large numbers of neurons. We apply these estimators to simulated and real neural data and show that they substantially outperform traditional methods.

## Introduction

Information theoretic quantities are popular tools in neuroscience, where they are used to study neural codes whose representation or function is unknown. Neuronal signals take the form of fast ($\sim 1$ ms) spikes which are frequently modeled as discrete, binary events. While the spiking response of even a single neuron has been the focus of intense research, modern experimental techniques make it possible to study the simultaneous activity of hundreds of neurons. At a given time, the response of a population of $n$ neurons may be represented by a binary vector of length $n$, where each entry represents the presence (1) or absence (0) of a spike. We refer to such a vector as a *spike word*. For $n$ much greater than 30, the space of $2^n$ spike words becomes so large that effective modeling and analysis of neural data, with their high dimensionality and relatively low sample size, presents a significant computational and theoretical challenge.

We study the problem of estimating the discrete entropy of spike word distributions. This is a difficult problem when the sample size is much less than $2^n$, the number of spike words. Entropy estimation in general is a well-studied problem with a literature spanning statistics, physics, neuro-

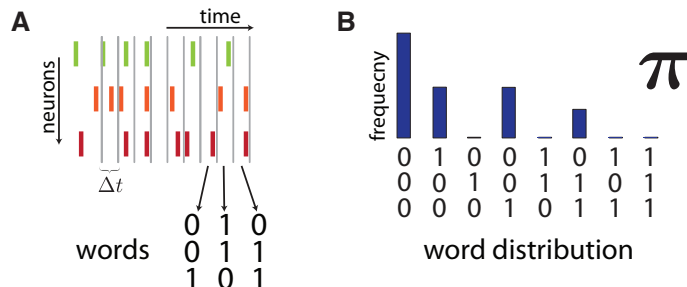

Figure 1: Illustrated example of binarized spike responses for $n = 3$ neurons and corresponding word distribution. **(A)** The spike responses of $n = 3$ simultaneously-recorded neurons (green, orange, and purple). Time is discretized into bins of size $\Delta t$. A single spike word is a $3 \times 1$ binary vector whose entries are 1 or 0 corresponding to whether the neuron spiked or not within the time bin. **(B)** We model spike words as drawn iid from the *word distribution* $\boldsymbol{\pi}$, a probability distribution supported on the $\mathcal{A} = 2^n$ unique binary words. Here we show a schematic $\boldsymbol{\pi}$ for the data of panel **(A)**. The spike words (x-axis) occur with varying probability (blue)

science, ecology, and engineering, among others [1–7]. We introduce a novel Bayesian estimator which, by incorporating simple *a priori* information about spike trains via a carefully-chosen prior, can estimate entropy with remarkable accuracy from few samples. Moreover, we exploit the structure of spike trains to compute efficiently on the full space of $2^n$ spike words.

We begin by briefly reviewing entropy estimation in general. In Section 2 we discuss the statistics of spike trains and emphasize a statistic, called the synchrony distribution, which we employ in our model. In Section 3 we introduce two novel estimators, the Dirichlet–Bernoulli (DBer) and Dirichlet–Synchrony (DSyn) entropy estimators, and discuss their properties and computation. We compare $\hat{H}_{\text{DBer}}$ and $\hat{H}_{\text{DSyn}}$ to other entropy estimation techniques in simulation and on neural data, and show that $\hat{H}_{\text{DBer}}$ drastically outperforms other popular techniques when applied to real neural data. Finally, we apply our estimators to study synergy across time of a single neuron.

## 1 Entropy Estimation

Let $\mathbf{x} := \{x_k\}_{k=1}^N$ be spike words drawn iid from an unknown word distribution $\boldsymbol{\pi} := \{\pi_i\}_{i=1}^{\mathcal{A}}$. There are $\mathcal{A} = 2^n$ unique words for a population of $n$ neurons, which we index by $\{1, 2, \ldots, \mathcal{A}\}$. Each sampled word $x_k$ is a binary vector of length $n$, where $x_{ki}$ records the presence or absence of a spike from the $i$th neuron. We wish to estimate the entropy of $\boldsymbol{\pi}$,

$$H(\boldsymbol{\pi}) = -\sum_{k=1}^{\mathcal{A}} \pi_k \log \pi_k, \tag{1}$$

where $\pi_k > 0$ denotes the probability of observing the $k$th word.

A naive method for estimating $H$ is to first estimate $\boldsymbol{\pi}$ using the count of observed words $n_k = \sum_{i=1}^{N} \mathbf{1}_{\{x_i = k\}}$ for each word $k$. This yields the empirical distribution $\hat{\boldsymbol{\pi}}$, where $\hat{\pi}_k = n_k/N$. Evaluating eq. 1 on this estimate yields the "plugin" estimator,

$$\hat{H}_{\text{plugin}} = -\sum_{i=1}^{\mathcal{A}} \hat{\pi}_i \log \hat{\pi}_i, \tag{2}$$

which is also the maximum-likelihood estimator under the multinomial likelihood. Although consistent and straightforward to compute, $\hat{H}_{\text{plugin}}$ is in general severely biased when $N \ll \mathcal{A}$.

Indeed, all entropy estimators are biased when $N \ll \mathcal{A}$ [8]. As a result, many techniques for bias-correction have been proposed in the literature [6, 9–18]. Here, we extend the Bayesian approach of [19], focusing in particular on the problem of entropy estimation for simultaneously-recorded neurons.

In a Bayesian paradigm, rather than attempting to directly compute and remove the bias for a given estimator, we instead choose a prior distribution over the space of discrete distributions. Nemenman

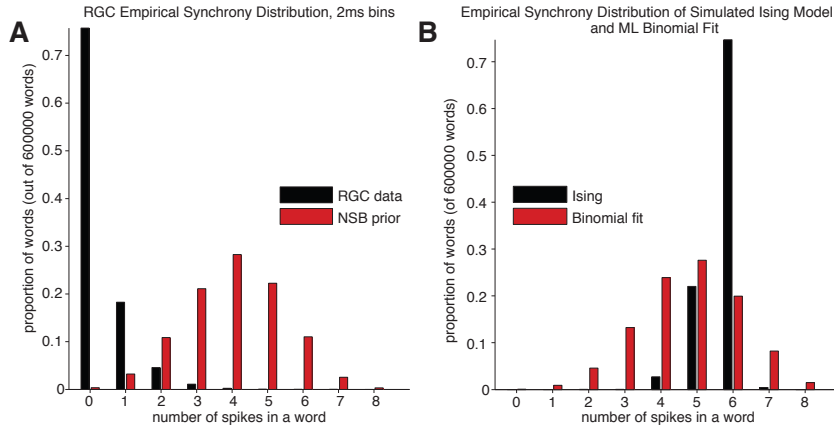

Figure 2: Sparsity structure of spike word distribution illustrated using the synchrony distribution. **(A)** The empirical synchrony distribution of $8$ simultaneously-recorded retinal ganglion cells (blue). The cells were recorded for $20$ minutes and binned with $\Delta t = 2$ ms bins. Spike words are overwhelmingly sparse, with $w_0$ by far the most common word. In contrast, we compare the prior empirical synchrony distribution sampled using $10^6$ samples from the NSB prior $(\boldsymbol{\pi} \sim \mathrm{Dir}(\alpha, \dots, \alpha))$, with $p(\alpha) \propto \mathcal{A}\psi_1(\mathcal{A}\alpha + 1) - \psi_1(\alpha + 1)$, and $\psi_1$ the digamma function) (red). The empirical synchrony distribution shown is averaged across samples. **(B)** The synchrony distribution of an Ising model (blue) compared to its best binomial fit (red). The Ising model parameters were set randomly by drawing the entries of the matrix $J$ and vector $h$ iid from $\mathcal{N}(0, 1)$. A binomial distribution cannot accurately capture the observed synchrony distribution.

*et al.* showed Dirichlet to be priors highly informative about the entropy, and thus a poor prior for Bayesian entropy estimation [19]. To rectify this problem, they introduced the Nemenman–Shafee–Bialek (NSB) estimator, which uses a mixture of Dirichlet distributions to obtain an approximately flat prior over $H$. As a prior on $\boldsymbol{\pi}$, however, the NSB prior is agnostic about application: all symbols have the same marginal probability under the prior, an assumption that may not hold when the symbols correspond to binary spike words.

## 2 Spike Statistics and the Synchrony Distribution

We discretize neural signals by binning multi-neuron spike trains in time, as illustrated in Fig. 1. At a time $t$, then, the spike response of a population of $n$ neurons is a binary vector $\vec{w} = (b_1, b_2, \dots, b_n)$, where $b_i \in \{0, 1\}$ corresponds to the event that the $i$th neuron spikes within the time window $(t, t + \Delta t)$. We let $\vec{w}_k$ be that word such that $k = \sum_{i=0}^{n-1} b_i 2^i$. There are a total of $\mathcal{A} = 2^n$ possible words.

For a sufficiently small bin size $\Delta t$, spike words are likely to be sparse, and so our strategy will be to choose priors that place high prior probability on sparse words. To quantify sparsity we use the *synchrony distribution*: the distribution of population spike counts across all words. In Fig. 2 we compare the empirical synchrony distribution for a population of $8$ simultaneously-recorded retinal ganglion cells (RGCs) with the prior synchrony distribution under the NSB model. For real data, the synchrony distribution is asymmetric and sparse, concentrating around words with few simultaneous spikes. No more than $4$ synchronous spikes are observed in the data. In contrast, under the NSB model all words are equally likely, and the prior synchrony distribution is symmetric and centered around $4$.

These deviations in the synchrony distribution are noteworthy: beyond quantifying sparseness, the synchrony distribution provides a surprisingly rich characterization of a neural population. Despite its simplicity, the synchrony distribution carries information about the higher-order correlation structure of a population [20, 21]. It uniquely specifies distributions $\boldsymbol{\pi}$ for which the probability of a word $w_k$ depends only on its spike count $[k] = [\vec{w}_k] := \sum_i b_i$. Equivalently: all words with spike count $k$, $\mathcal{E}_k = \{w : [w] = k\}$, have identical probability $\beta_k$ of occurring. For such a $\boldsymbol{\pi}$, the synchrony

distribution $\mu$ is given by,

$$\mu_k = \sum_{w_i \in \mathcal{E}_k} \pi_i = \binom{n}{k}\beta_k. \tag{3}$$

Different neural models correspond to different synchrony distributions. Consider an independently-Bernoulli spiking model. Under this model, the number of spikes in a word $w$ is distributed binomially, $[\vec{w}] \sim \mathrm{Bin}(p, n)$, where $p$ is a uniform spike probability across neurons. The probability of a word $w_k$ is given by,

$$P(\vec{w}_k|p) = \beta_{[k]} = p^{[k]}(1-p)^{n-[k]}, \tag{4}$$

while the probability of observing a word with $i$ spikes is,

$$P(\mathcal{E}_i|p) = \binom{n}{i}\beta_i. \tag{5}$$

## 3 Entropy Estimation with Parametric Prior Knowledge

Although a synchrony distribution may capture our prior knowledge about the structure of spike patterns, our goal is not to estimate the synchrony distribution itself. Rather, we use it to inform a prior on the space of discrete distributions, the $(2^n-1)$-dimensional simplex. Our strategy is to use a synchrony distribution $G$ as the *base measure* of a Dirichlet distribution. We construct a hierarchical model where $\boldsymbol{\pi}$ is a mixture of $\mathrm{Dir}(\alpha G)$, and counts $\mathbf{n}$ of spike train observations are multinomial given $\boldsymbol{\pi}$ (See Fig. 3(**A**). Exploiting the conjugacy of Dirichlet and multinomial, and the convenient symmetries of both the Dirichlet distribution and $G$, we obtain a computationally efficient Bayes least squares estimator for entropy. Finally, we discuss using empirical estimates of the synchrony distribution $\boldsymbol{\mu}$ as a base measure.

### 3.1 Dirichlet–Bernoulli entropy estimator

We model spike word counts $\mathbf{n}$ as drawn iid multinomial given the spike word distribution $\boldsymbol{\pi}$. We place a mixture-of-Dirichlets prior on $\boldsymbol{\pi}$, which in general takes the form,

$$\mathbf{n} \sim \mathrm{Mult}(\boldsymbol{\pi}) \tag{6}$$

$$\boldsymbol{\pi} \sim \mathrm{Dir}(\underbrace{\alpha_1, \alpha_2, \ldots, \alpha_{\mathcal{A}}}_{2^n}), \tag{7}$$

$$\vec{\alpha} := (\alpha_1, \alpha_2, \ldots, \alpha_{\mathcal{A}}) \sim P(\vec{\alpha}), \tag{8}$$

where $\alpha_i > 0$ are *concentration* parameters, and $P(\vec{\alpha})$ is a prior distribution of our choosing. Due to the conjugacy of Dirichlet and multinomial, the posterior distribution given observations and $\vec{\alpha}$ is $\boldsymbol{\pi}|\mathbf{n}, \vec{\alpha} \sim \mathrm{Dir}(\alpha_1 + n_1, \ldots, \alpha_{\mathcal{A}} + n_{\mathcal{A}})$, where $n_i$ is the number of observations for the $i$-th spiking pattern. The posterior expected entropy given $\vec{\alpha}$ is given by [22],

$$\mathbb{E}[H(\boldsymbol{\pi})|\vec{\alpha}] = \psi_0(\kappa + 1) - \sum_{i=1}^{\mathcal{A}} \frac{\alpha_i}{\kappa} \psi_0(\alpha_i + 1) \tag{9}$$

where $\psi_0$ is the digamma function, and $\kappa = \sum_{i=1}^{\mathcal{A}} \alpha_i$.

For large $\mathcal{A}$, $\vec{\alpha}$ is too large to select arbitrarily, and so in practice we *center* the Dirichlet around a simple, parametric base measure $G$ [23]. We rewrite the vector of concentration parameters as $\vec{\alpha} \equiv \alpha G$, where $G = \mathrm{Bernoulli}(p)$ is a Bernoulli distribution with spike rate $p$ and $\alpha > 0$ is a scalar. The general prior of eq. 7 then takes the form,

$$\boldsymbol{\pi} \sim \mathrm{Dir}(\alpha G) \equiv \mathrm{Dir}(\alpha g_1, \alpha g_2 \ldots, \alpha g_{\mathcal{A}}), \tag{10}$$

where each $g_k$ is the probability of the $k$th word under the base measure, satisfying $\sum g_k = 1$.

We illustrate the dependency structure of this model schematically in Fig. 3. Intuitively, the base measure incorporates the structure of $G$ into the prior by shifting the Dirichlet's mean. With a base measure $G$ the prior mean satisfies $\mathbb{E}[\boldsymbol{\pi}|p] = G|p$. Under the NSB model, $G$ is the uniform distribution; thus, when $p = 0.5$ the Binomial $G$ corresponds exactly to the NSB model. Since

in practice choosing a base measure is equivalent to selecting distinct values of the concentration parameter $\alpha_i$, the posterior mean of entropy under this model has the same form as eq. 9, simply replacing $\alpha_k = \alpha g_k$. Given hyper-prior distributions $P(\alpha)$ and $P(p)$, we obtain the Bayes least squares estimate, the posterior mean of entropy under our model,

$$\hat{H}_{\text{DBer}} = \mathbb{E}[H|\mathbf{x}] = \iint \mathbb{E}\left[H|\alpha, p\right] P(\alpha, p|\mathbf{x}) \, \mathrm{d}\alpha \, \mathrm{d}p. \tag{11}$$

We refer to eq. 11 as the Dirichlet–Bernoulli (DBer) entropy estimator, $\hat{H}_{\text{DBer}}$. Thanks to the closed-form expression for the conditional mean eq. 9 and the convenient symmetries of the Bernoulli distribution, the estimator is fast to compute using a 2D numerical integral over the hyperparameters $\alpha$ and $p$.

### 3.1.1 Hyper-priors on $\alpha$ and $p$

Previous work on Bayesian entropy estimation has focused on Dirichlet priors with scalar, constant concentration parameters $\alpha_i = \alpha$. Nemenman *et al.* [19] noted that these fixed-$\alpha$ priors yield poor estimators for entropy, because $p(H|\alpha)$ is highly concentrated around its mean. To address this problem, [19] proposed a Dirichlet mixture prior on $\boldsymbol{\pi}$,

$$P(\boldsymbol{\pi}) = \int P_{\text{Dir}}(\boldsymbol{\pi}|\alpha)P(\alpha)\mathrm{d}\alpha, \tag{12}$$

where the hyper-prior, $P(\alpha) \propto \frac{\mathrm{d}}{\mathrm{d}\alpha}\mathbb{E}[H(\boldsymbol{\pi})|\alpha]$ assures an approximately flat prior distribution over $H$. We adopt the same strategy here, choosing the prior,

$$P(\alpha) \propto \frac{\mathrm{d}}{\mathrm{d}\alpha}\mathbb{E}[H(\boldsymbol{\pi})|\alpha, p] = \psi_1(\alpha + 1) - \sum_{i=0}^{n}\binom{n}{i}\beta_i^2\psi_1(\alpha\beta_i + 1). \tag{13}$$

Entropy estimates are less sensitive to the choice of prior on $p$. Although we experimented with several priors on $p$, in all examples we found that the evidence for $p$ was highly concentrated around $\hat{p} = \frac{1}{Nn}\sum_{ij} x_{ij}$, the maximum (Bernoulli) likelihood estimate for $p$. In practice, we found that an empirical Bayes procedure, fitting $\hat{p}$ from data first and then using the fixed $\hat{p}$ to perform the integral eq. 11, performed indistinguishably from a $P(p)$ uniform on $[0, 1]$.

### 3.1.2 Computation

For large $n$, the $2^n$ distinct values of $\alpha_i$ render the sum of eq. 9 potentially intractable to compute. We sidestep this exponential scaling of terms by exploiting the redundancy of Bernoulli and binomial distributions. Doing so, we are able to compute eq. 9 without explicitly representing the $2^N$ values of $\alpha_i$.

Under the Bernoulli model, each element $g_k$ of the base measure takes the value $\beta_{[k]}$ (eq. 4). Further, there are $\binom{n}{i}$ words for which the value of $\alpha_i$ is identical, so that $A = \sum_{i=0}^{n}\alpha\binom{n}{i}\beta_i = \alpha$. Applied to eq. 9, we have,

$$\mathbb{E}[H(\boldsymbol{\pi})|\alpha, p] = \psi_0(\alpha + 1) - \sum_{i=0}^{n}\binom{n}{i}\beta_i\psi_0(\alpha\beta_i + 1).$$

For the posterior, the sum takes the same form, except that $A = n + \alpha$, and the mean is given by,

$$\mathbb{E}[H(\boldsymbol{\pi})|\alpha, p, \mathbf{x}] = \psi_0(n + \alpha + 1) - \sum_{i=1}^{\mathcal{A}} \frac{n_i + \alpha\beta_{[i]}}{n + \alpha}\psi_0(n_i + \alpha\beta_{[i]} + 1) \tag{14}$$

$$= \psi_0(n + \alpha + 1) - \sum_{i \in \mathcal{I}} \frac{n_i + \alpha\beta_{[i]}}{n + \alpha}\psi_0(n_i + \alpha\beta_{[i]} + 1)$$

$$- \alpha\sum_{i=0}^{n} \frac{\left(\binom{n}{i} - \tilde{n}_i\right)\beta_i}{n + \alpha}\psi_0(\alpha\beta_i + 1),$$

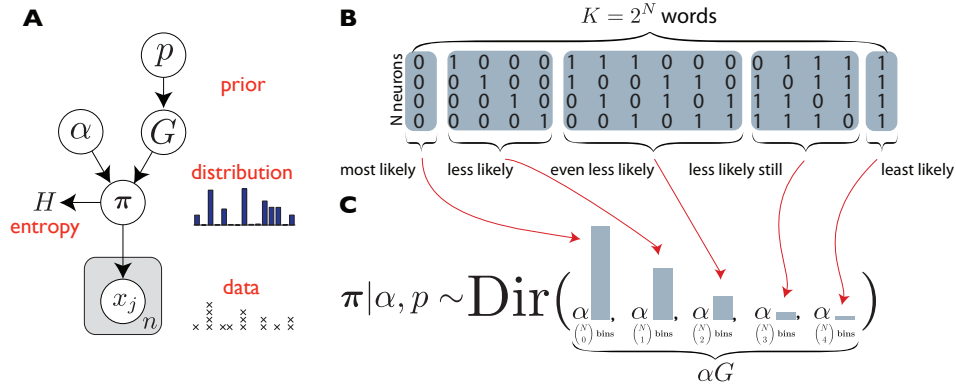

Figure 3: Model schematic and intuition for Dirichlet–Bernoulli entropy estimation. **(A)** Graphical model for Dirichlet–Bernoulli entropy estimation. The Bernoulli base measure $G$ depends on the spike rate parameter $p$. In turn, $G$ acts as the mean of a Dirichlet prior over $\pi$. The scalar Dirichlet concentration parameter $\alpha$ determines the variability of the prior around the base measure. **(B)** The set of possible spike words for $n = 4$ neurons. Although easy to enumerate for this small special case, the number of words increases exponentially with $n$. In order to compute with this large set, we assume a prior distribution with a simple equivalence class structure: *a priori*, all words with the same number of synchronous spikes (outlined in blue) occur with equal probability. We then need only $n$ parameters, the synchrony distribution of eq. 3, to specify the distribution. **(C)** We center a Dirichlet distribution on a model of the synchrony distribution. The symmetries of the count and Dirichlet distributions allow us to compute without explicitly representing all $\mathcal{A}$ words.

where $\mathcal{I} = \{k : n_k > 0\}$, the set of observed characters, and $\tilde{n}_i$ is the count of observed words with $i$ spikes (i.e., observations of the equivalence class $\mathcal{E}_i$). Note that eq. 14 is much more computationally tractable than the mathematically equivalent form given immediately above it. Thus, careful bookkeeping allows us to efficiently evaluate eq. 9 and, in turn, eq. 11.[1]

### 3.2 Empirical Synchrony Distribution as a Base Measure

While the Bernoulli base measure captures the sparsity structure of multi-neuron recordings, it also imposes unrealistic independence assumptions. In general, the synchrony distribution can capture correlation structure that cannot be represented by a Bernoulli model. For example, in Fig. 2**B**, a maximum likelihood Bernoulli fit fails to capture the sparsity structure of a simulated Ising model.

We might address this by choosing a more flexible parametric base measure. However, since the dimensionality of $\mu$ scales only linearly with the number of neurons, the empirical synchrony distribution (ESD),

$$\hat{\mu}_i = \frac{1}{N} \sum_{j=1}^{N} \mathbf{1}_{\{[x_j]=i\}}, \tag{15}$$

converges quickly even when the sample size is inadequate for estimating the full $\pi$.

This suggests an empirical Bayes procedure where we use the ESD as a base measure (take $G = \hat{\mu}$) for entropy estimation. Computation proceeds exactly as in Section 3.1.2 with the Bernoulli base measure replaced by the ESD by setting $g_k = \mu_k$ and $\beta_i = \mu_i/\binom{m}{i}$. The resulting Dirichlet–Synchrony (DSyn) estimator may incorporate more varied sparsity and correlation structures into its prior than $\hat{H}_{\text{DBer}}$ (see Fig. 4), although it depends on an estimate of the synchrony distribution.

## 4 Simulations and Comparisons

We compared $\hat{H}_{\text{DBer}}$ and $\hat{H}_{\text{DSyn}}$ to the Nemenman–Shafee–Bialek (NSB) [19] and Best Upper Bound (BUB) entropy estimators [8] for several simulated and real neural datasets. For $\hat{H}_{\text{DSyn}}$, we regular-

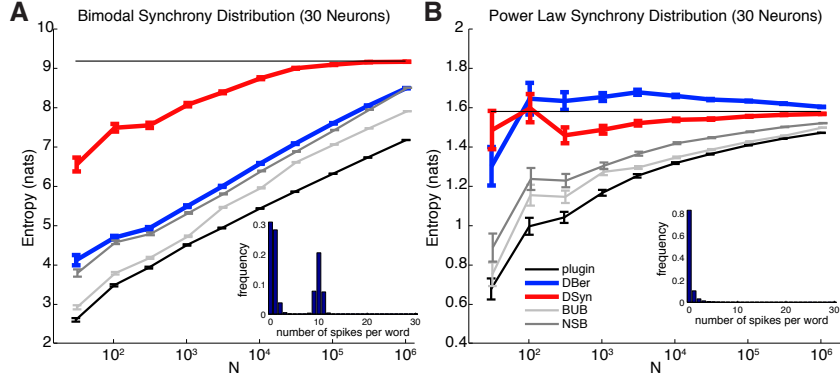

Figure 4: Convergence of $\hat{H}_{\text{DBer}}$, $\hat{H}_{\text{DSyn}}$, $\hat{H}_{\text{NSB}}$, $\hat{H}_{\text{BUB}}$, and $\hat{H}_{\text{plugin}}$ as a function of sample size for two simulated examples of 30 neurons. Binary word data are drawn from two specified synchrony distributions (insets). Error bars indicate variability of the estimator over independent samples ($\pm 1$ standard deviation). **(A)** Data drawn from a bimodal synchrony distribution with peaks at 0 spikes and 10 spikes $\left( \mu_i = e^{-2i} + \frac{1}{10} e^{-4(i-2n/3)^2} \right)$. **(B)** Data generated from a power-law synchrony distribution ($\mu_i \propto i^{-3}$).

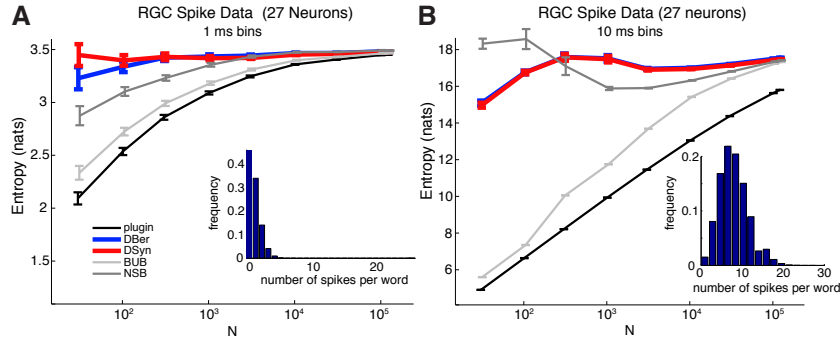

Figure 5: Convergence of $\hat{H}_{\text{DBer}}$, $\hat{H}_{\text{DSyn}}$, $\hat{H}_{\text{NSB}}$, $\hat{H}_{\text{BUB}}$, and $\hat{H}_{\text{plugin}}$ as a function of sample size for 27 simultaneously-recorded retinal ganglion cells (RGC). The two figures show the same RGC data binned and binarized at $\Delta t = 1$ ms **(A)** and 10 ms **(B)**. The error bars, axes, and color scheme are as in Fig. 4. While all estimators improve upon the performance of $\hat{H}_{\text{plugin}}$, $\hat{H}_{\text{DSyn}}$ and $\hat{H}_{\text{DBer}}$ both show excellent performance for very low sample sizes (10's of samples). **(inset)** The empirical synchrony distribution estimated from 120 minutes of data.

ized the estimated ESD by adding a pseudo-count of $\frac{1}{K}$, where $K$ is the number of unique words observed in the sample. In Fig. 4 we simulated data from two distinct synchrony distribution models. As is expected, among all estimators, $\hat{H}_{\text{DSyn}}$ converges the fastest with increasing sample size $N$. The $\hat{H}_{\text{DBer}}$ estimator converges more slowly, as the Bernoulli base measure is not capable of capturing the correlation structure of the simulated synchrony distributions. In Fig. 5, we show convergence performance on increasing subsamples of 27 simultaneously-recorded retinal ganglion cells. Again, $\hat{H}_{\text{DBer}}$ and $\hat{H}_{\text{DSyn}}$ show excellent performance. Although the true word distribution is not described by a synchrony distribution, the ESD proves an excellent regularizer for the space of distributions, even for very small sample sizes.

## 5 Application: Quantification of Temporal Dependence

We can gain insight into the coding of a single neural time-series by quantifying the amount of information a single time bin contains about another. The correlation function (Fig. 6**A**) is the statistic most widely used for this purpose. However, correlation cannot capture higher-order dependencies. In neuroscience, mutual information is used to quantify higher-order temporal structure [24]. A re-

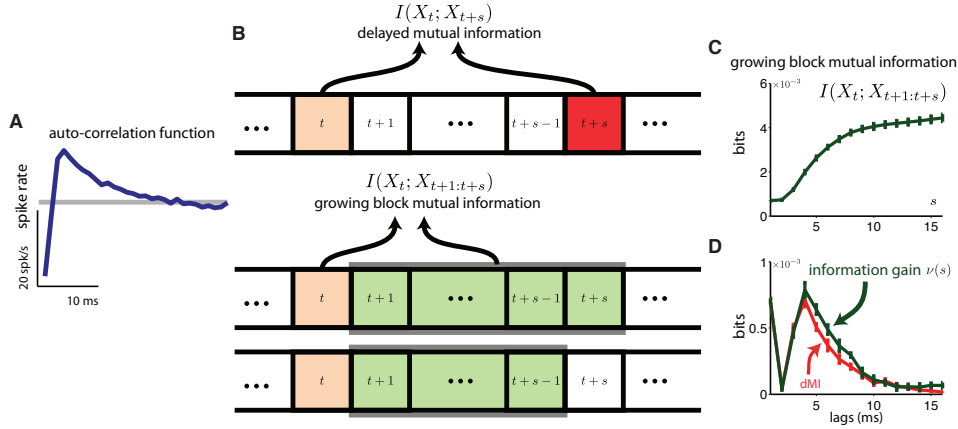

Figure 6: Quantifying temporal dependence of RGC coding using $\hat{H}_{\text{DBer}}$. **(A)** The auto-correlation function of a single retinal ganglion neuron. Correlation does not capture the full temporal dependence. We bin with $\Delta t = 1$ ms bins. **(B)** Schematic definition of time delayed mutual information (dMI), and block mutual information. The information gain of the $s$th bin is $\nu(s) = I(X_t; X_{t+1:t+s}) - I(X_t; X_{t+1:t+s-1})$. **(C)** Block mutual information estimate as a function of growing block size. Note that the estimate is monotonically increasing, as expected, since adding new bins can only increase the mutual information. **(D)** Information gain per bin assuming temporal independence (dMI), and with difference between block mutual informations ($\nu(s)$). We observe synergy for the time bins in the $5$ to $10$ ms range.

lated quantity, the delayed mutual information (dMI) provides an indication of instantaneous dependence: $dMI(s) = I(X_t; X_{t+s})$, where $X_t$ is a binned spike train, and $I(X; Y) = H(X) - H(X|Y)$ denotes the mutual information. However, this quantity ignores any temporal dependences in the intervening times: $X_{t+1}, \ldots, X_{t+s-1}$. An alternative approach allows us to consider such dependences: the "block mutual information" $\nu(s) = I(X_t; X_{t+1:t+s}) - I(X_t; X_{t+1:t+s-1})$ (Fig. 6**B**,**C**,**D**)

The relationship between $\nu(s)$ and $dMI(s)$ provides insight about the information contained in the recent history of the signal. If each time bin is conditionally independent given $X_t$, then $\nu(s) = dMI(s)$. In contrast, if $\nu(s) < dMI(s)$, instantaneous dependence is partially explained by history. Finally, $\nu(s) > dMI(s)$ implies that the joint distribution of $X_t, X_{t+1}, \ldots, X_{t+s}$ contains more information about $X_t$ than the joint distribution of $X_t$ and $X_{t+s}$ alone. We use the $\hat{H}_{\text{DBer}}$ entropy estimator to compute mutual information (by computing $H(X)$ and $H(X|Y)$) accurately for $\sim 15$ bins of history. Surprisingly, individual retinal ganglion cells code synergistically in time (Fig. 6**D**).

## 6 Conclusions

We introduced two novel Bayesian entropy estimators, $\hat{H}_{\text{DBer}}$ and $\hat{H}_{\text{DSyn}}$. These estimators use a hierarchical mixture-of-Dirichlets prior with a base measure designed to integrate *a priori* knowledge about spike trains into the model. By choosing base measures with convenient symmetries, we simultaneously sidestepped potentially intractable computations in the high-dimensional space of spike words. It remains to be seen whether these symmetries, as exemplified in the structure of the synchrony distribution, are applicable across a wide range of neural data. Finally, however, we showed several examples in which these estimators, especially $\hat{H}_{\text{DSyn}}$, perform exceptionally well in application to neural data. A MATLAB implementation of the estimators will be made available at https://github.com/pillowlab/CDMentropy.

## Acknowledgments

We thank E. J. Chichilnisky, A. M. Litke, A. Sher and J. Shlens for retinal data. This work was supported by a Sloan Research Fellowship, McKnight Scholar's Award, and NSF CAREER Award IIS-1150186 (JP).

## Footnotes

[1]For large $n$, the binomial coefficient of eq. 14 may be difficult to compute. By writing it in terms of the Bernoulli probability eq. 5, it may be computed using the Normal approximation to the Binomial.

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
