[Reviews · NeurIPS 2013]

Submitted by Assigned_Reviewer_5

As far as I can see, the method favors descriptions which are mainly based on the spike count. I think, it would be interesting to see, how the method performs in situations, in which there are other higher-order dependencies apart from spike count. Specifically, including a comparison between other methods such as BUB in situations, in which spike-trains are, for example, generated according to a GLM, would be a more convincing argument of the accuracy.

minor points:
line 163: reference missing
line 221: \alpha=0 implies uniform distribution?
line 296: for -->form
Summary: The manuscript is well written and the feasibility of the approach is sufficiently demonstrated by the presented experiments.

Submitted by Assigned_Reviewer_6

Paper summary:

The paper describes two novel entropy estimators for binary neural spike words. The estimators are Bayesian and make use of a mixture-of-Dirichlet prior. The distribution is hierarchical with a count distribution as the base measure of the Dirichlet distribution. The authors evaluate their methods on artificial data and on data recorded simultaneously from retinal ganglion cells and compare them to established entropy estimators. They show that their estimators need less samples for accurate estimations. Finally, they apply their method to quantify temporal synergy in retinal ganglion cell data.


Quality:

The Bayesian entropy estimators are powerful, elegantly evading the curse of dimensionality. By including prior information about the structure of the problem, the method reduces the number of required samples.

On the synthetic data, it is not surprising that the proposed estimators outperform the alternative estimators, since the distributions of the word data follow the model structure of the estimators. The performance on the data recorded simultaneously from retinal ganglion cells is impressive.

It is not clear, though, how well the methods would do on other neural data. The method contains the critical underlying assumption that the word distribution is well characterized by the overall count distribution. For the retinal ganglion cell data this is apparently the case, but further evaluations will have to show whether or not this will also hold in general. It might be worth to mention this problem with a sentence in the discussion. In any case, the new estimators are certainly very useful.


Clarity:

The paper is nicely written.


Originality:

The proposed entropy estimators extend the work by Nemenman et al., NIPS 2002 by including prior knowledge about the structure of the spike trains. The general idea is similar to that of the raster marginals model (Okun et al., J Neurosci 2012) in that the total spike count distribution is used as a simplification to evade the curse of dimensionality.


Significance:

Entropy estimation is a very important problem, because information quantification is a central problem of neural coding analyses. The demonstrated performance gain compared to alternative methods is impressive.


Minor points:
080: In Section 3 introduce -> In Section 3, we introduce
163: Citation missing
232: Incomplete sentence
234: the estimator fast-to-compute -> the estimator is fast-to-compute
290: We only then need only -> We then need only
295: for -> form
Figures 4 and 5: DCnt -> DCt
Summary: The paper introduces entropy estimators for neural spike trains that require less samples for accurate estimations. The contribution is important and well implemented.

Submitted by Assigned_Reviewer_7

This submission presents a straightforward improvement on current estimators of entropy in multidimensional binary distributions. Estimating entropy (and related measures like mutual information) is considered important in advancing our understanding of the neural code. This work is especially timely in that an increasing number of multi-electrode recordings are currently taking place whose interpretability depends on better analysis methods like the one presented here.

The work presented here relies on the fact that population responses, esp. when binned in short time intervals, are very sparse: the most frequent word is one in which no neuron spikes, with a word frequency rapidly decreasing with the number of spikes, or ones, in a word. By incorporating this prior knowledge into their Bayesian estimator, the authors derive an estimator that achieve a bias comparable to that of existing estimators using several orders of magnitude less data - a critical constraint in empirical studies.

In an example application of their method to real data, the authors find that retinal ganglion cells "code synergistically" in time. While interesting, this section would need to be expanded to yield robust and convincing insights.

The presentation of the derivation and the results is quite clear and I only have minor comments.
[deleted]
Summary: Important improvement on entropy estimation methods based on a more sophisticated prior than used previously.
Author Feedback

Author rebuttal: We thank all the reviewers for their careful reading of our work, and
for their many thoughtful comments and helpful suggestions.

Reviewer 5:

Thank you for the excellent suggestion regarding simulation from a GLM. We will attempt this for the revision.

Reviewer 6:

We certainly agree that performance of our estimator will depend upon the extent to which the count distribution is capable of describing the full joint distribution of words, and we will clarify this point in the discussion. We note, however, that so long as the count distribution captures any aspect of the joint that differs from uniform, we expect the DCt estimator to have an advantage over NSB (i.e., since the latter effectively models all spike patterns as equally likely a priori, whereas DCt will capture any tendency for the count distribution to differ from a binomial centered at N/2).

Reviewer 7:

Although the current manuscript only shows results for the case of small bin size, where spikes are sparse, further experiments on the same RGC data with larger bin sizes show that the method also performs well on less-sparse data. We will include an additional figure showing this.

We agree that our application section would benefit from expansion. The goal of the analyses shown in Fig 6. was to demonstrate an application of entropy estimation to a scientific question, which displaced an additional comparison between entropy estimators on neural data. We will explore applications further in future work.